molecular biology/biochemistry/cellular biology

replication stress, common fragile sites, mitotic DNA repair synthesis, break-induced replication, ultrafine DNA bridges, chromosome instability

**Author for correspondence:**
Kok-Lung Chan
e-mail: koklung.chan@sussex.ac.uk

†Present address: The Institute of Cancer Research, Chester Beatty Laboratories, 237 Fulham Rd, Chelsea, London SW3 6JB, UK.

One contribution to Life Sciences New Talent special collection.

# Mind the replication gap

## Camelia Mocanu† and Kok-Lung Chan

Chromosome Dynamics and Stability Group, Genome Damage and Stability Centre, University of Sussex, Brighton BN1 7BG, UK

K-LC, 0000-0002-8796-8385

Unlike bacteria, mammalian cells need to complete DNA replication before segregating their chromosomes for the maintenance of genome integrity. Thus, cells have evolved efficient pathways to restore stalled and/or collapsed replication forks during S-phase, and when necessary, also to delay cell cycle progression to ensure replication completion. However, strong evidence shows that cells can proceed to mitosis with incompletely replicated DNA when under mild replication stress (RS) conditions. Consequently, the incompletely replicated genomic gaps form, predominantly at common fragile site regions, where the converging fork-like DNA structures accumulate. These branched structures pose a severe threat to the faithful disjunction of chromosomes as they physically interlink the partially duplicated sister chromatids. In this review, we provide an overview discussing how cells respond and deal with the under-replicated DNA structures that escape from the S/G2 surveillance system. We also focus on recent research of a mitotic break-induced replication pathway (also known as mitotic DNA repair synthesis), which has been proposed to operate during prophase in an attempt to finish DNA synthesis at the under-replicated genomic regions. Finally, we discuss recent data on how mild RS may cause chromosome instability and mutations that accelerate cancer genome evolution.

# 1. Replication stress alters the timing of cell cycle progression

DNA replication and cell division are crucial for the continuity of life. These two events are tightly coordinated and regulated throughout the cell cycle to ensure faithful transmission of genetic material to every offspring cell. Generally, the cell cycle is divided into four stages: Gap period 1/0 (G1/0), Synthesis phase (S), Gap period 2 (G2) and Mitosis (M). Cells that are in a non-proliferating stage are usually referred to as G0 phase, whereas G1 phase marks the beginning of the cell division cycle, in which two critical events for the initiation of DNA replication take place; determination and licensing of replication origins. Upon the stimulation of growth signalling and silencing of the G1/S DNA damage checkpoint, DNA replication

commences at the licensed origins via the activation of CMG (Cdc45-Mcm2/7-GINS) complex and replicative DNA polymerases [1]. The initiation of DNA synthesis does not necessarily happen at every licensed origin, and those without activation are generally termed as 'dormant origins' [2]. These dormant origins are believed to serve as backup sites for DNA synthesis re-initiation when normal replication progression is disrupted [3]. Following the S-phase is a second gap period called G2, where increased expression and activity of the pro-mitotic factors and kinases, such as cyclin-dependent kinase 1 (CDK1) and polo-like kinase 1 (PLK1), are observed. When the activities of these factors reach critical levels, it triggers mitotic entry or M-phase [4], in which the duplicated DNA molecules undergo a dramatic change to form a high-order chromosome structure via the action of condensin complexes—a process called chromosome condensation [5]. This facilitates the subsequent chromosome alignment and the separation of sister chromatids in the later stages of mitosis. After mitotic exit, each daughter cell will receive an identical set of fully duplicated chromosomes. The offspring cells will either undergo another round of the cell cycle or enter the non-proliferative G0 stage [6–8].

Given that proper segregation of chromosomes is hampered by the presence of incompletely replicated DNA structures, it is reasonable to think that crosstalk between DNA replication and mitosis might exist. Recently, evidence showing how S-phase and mitosis are coupled has emerged (figure 1). The activation of CDK1 and PLK1 mitotic kinases has been observed to start at the end of bulk genome duplication in late S-phase, which implicates a possible link between DNA replication and G2 (and mitosis) onset [9]. Indeed, a recent study demonstrates that DNA replication activity directly affects the timing of G2/M progression. Partial suppression of DNA synthesis activity increases the duration of the S-phase and hence delays mitosis onset. However, if a complete abrogation of DNA replication is imposed experimentally by conditional degradation of CDC6 and chemical inhibition of CDC7, it advances mitotic initiation [10]. Mechanistically, CDK2 activity increases during S-phase, promoting both DNA replication and the activity of mitotic kinases CDK1 and PLK1 [10]. However, the ongoing DNA replication activity can simultaneously suppress CDK1 and PLK1 activities presumably owing to the formation of single-stranded DNA structures and stochastic DNA damage that activate checkpoint kinases such as CHK1 [11]. This 'Yin and Yang' co-action is believed to fine tune the mitotic kinase activities once S-phase commences, and to ensure the timely initiation of mitosis only after DNA replication. Therefore, premature entry of mitosis is restricted [10]. Along this line, accumulating evidence has revealed that in budding yeast [12], avian cells [13] and more recently in human cells [14], ATR (Ataxia telangiectasia mutated and Rad3-related) kinase, in fact, acts as a critical coordinator between DNA replication and mitosis. Daigh *et al.* show that under replication stress (RS) conditions, ATR inhibits CDK2, hence downregulates global origin firing, and prolongs S-phase duration [14]. Furthermore, ATR also restricts the activity of CDK1 and PLK1 through phosphorylation of its downstream mediator, CHK1 [10]. Consequently, mitotic onset is delayed. Interestingly, the Cimprich group demonstrated that ATR also couples DNA replication with G2-phase onset by preventing premature CDK1-mediated phosphorylation on the FOXM1 transcription factor, which drives the transcription of pro-mitotic network genes such as cyclin B1 for mitotic onset [15]. Therefore, this work shows how S-phase kinases influence the mitosis-promoting factors, and explains why the lack of ATR activity leads to premature mitotic entry despite the presence of ongoing DNA synthesis activity [13]. Importantly, this strengthens the importance of separating DNA replication from mitosis; otherwise, severe chromosome under-condensation, malformation and DNA lesions can be introduced [2,12,13].

In unperturbed conditions, cells are usually able to fully duplicate their entire genome before mitosis. However, endogenous and exogenous DNA insults, or genetic defects in the DNA replication and/or repair machinery, can interfere with normal DNA synthesis, affecting its completion and the dynamics of the cell cycle [16]. This results in the loss of coordination between S-phase and mitosis. In this review, we will focus on discussing how cells cope with a situation of (mild) RS—a condition where replication fork movement is heavily slowed down, preventing DNA replication completion. The causes of RS are multifaceted and among the very well-studied ones include: aberrant origin firing [17,18], the clash between the replication and transcription machinery [19,20], deprivation of the building blocks for DNA metabolism and inhibition of replicative DNA polymerases [21,22]. In (pre)cancerous cells, one of the underlying causes of RS is believed to be proto-oncogene overexpression (e.g. c-MYC, h-RAS, cyclin E) [23]. Alternatively, RS can be induced as a result of nucleotide pool deficiency. Hydroxyurea (HU) is a commonly used chemical to deplete dNTP pools. It acts through the inhibition of ribonucleotide reductase (RNR) [24]. Treatments of specific DNA polymerase inhibitors also induce RS. A very well-documented one is aphidicolin (APH), which

*(a)*    (i) normal cell cycle progression

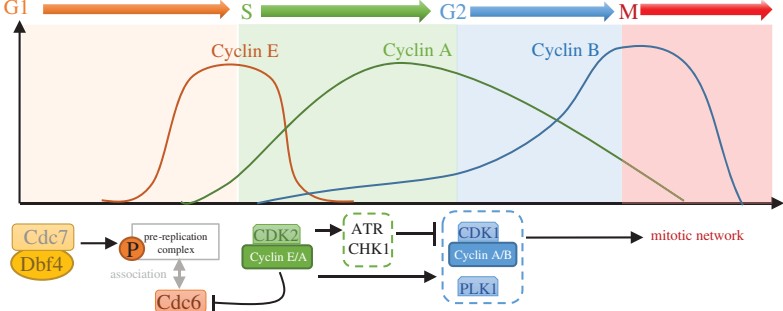

(ii) prolonged S-phase during replication stress

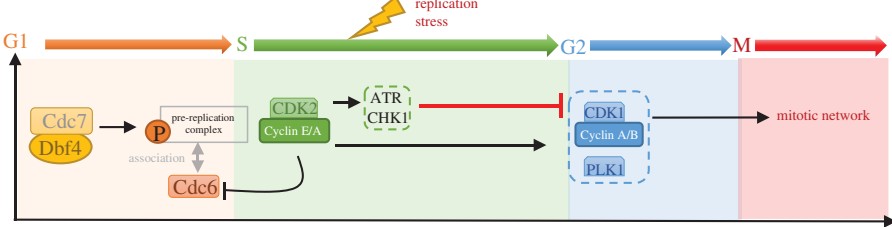

(iii) accelerated entry into mitosis when cdc6 and cdc7 function is lost

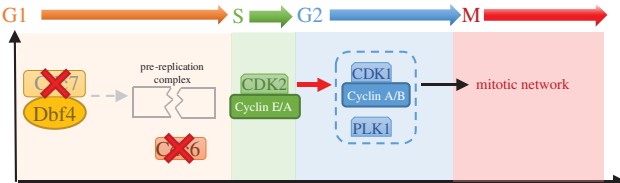

*(b)*

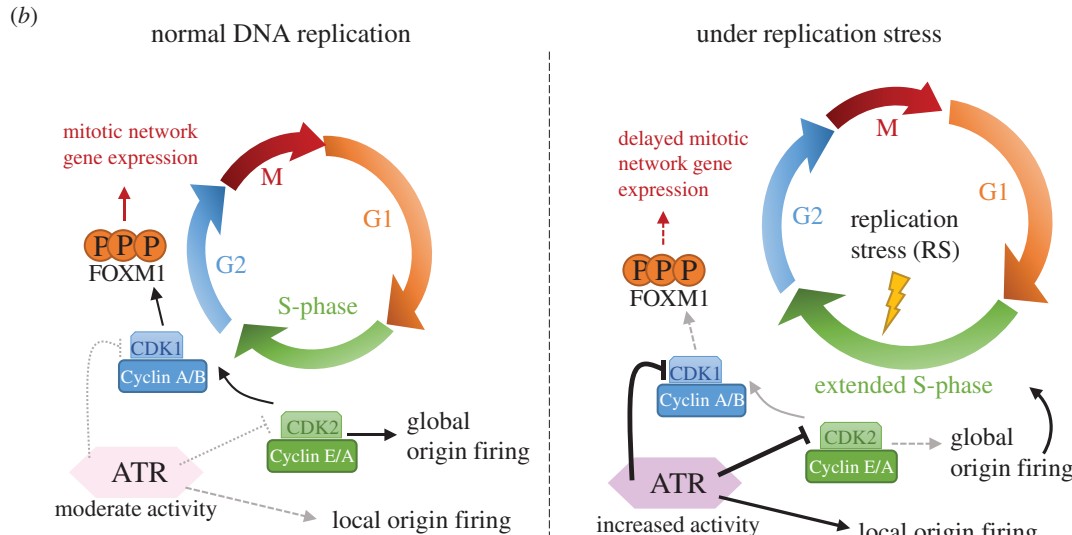

**Figure 1.** (Caption overleaf.)

inhibits the B-family of replicative DNA polymerases [24]. Under high doses, APH can stop DNA synthesis and alters origin firing [25]; however, when it is applied in sub-micromolar concentrations, it slows down DNA synthesis rates. In principle, slowing down replisome progression can increase the chance of a collision between replication and transcription machinery, elevating R-loop formation, a type of three-stranded structure formed by DNA:RNA hybrids [19,26]. Moreover, it has been also shown that in addition to the direct effect on DNA synthesis, RS also negatively impacts mitosis, presumably through the accumulation of replication and recombination structures. Well-known phenotypes include the formation of chromosomal gaps and/or breaks, multipolar spindles, and bulky and ultrafine anaphase DNA bridges. Therefore, a strict coordination between S-phase and

**Figure 1.** (*Overleaf.*) ATR coordinates S-phase progression and G2/mitotic entry. (*a*) Transition between different cell cycle phases is dependent on the activity of cyclins and cyclin-dependent kinase complexes. CDK2 associates with cyclin E during G1 and early S-phase, together with Dbf4-dependent kinase (DDK), Cdc7, they ensure the assembly of pre-replication complexes, which primes the chromatin regions for DNA replication initiation. The CDK2-cyclin A complex is responsible for initiation and completion of DNA synthesis. During late S-phase, CDK1 is also activated by binding to cyclin A, and later by cyclin B, which drives transcription signals to activate the mitotic gene network. During unperturbed conditions, the pre-replication complex (pre-RC) is assembled on chromatin. In G1/S, Cdc6 binds the pre-RC, leading to the recruitment of factors required for licensing of DNA replication (not shown). The Cdc7/Dbf4 kinase complex phosphorylates the pre-RC, allowing for DNA replication to start. CDK2-cyclin E/A complex represses Cdc6 to re-associate with the origins during S-phase, which hence prevents DNA re-replication. Furthermore, this kinase complex also promotes ATR-Chk1, CDK1 and PLK1 activity. As the ATR-Chk1 signalling pathway represses CDK1 activity, a feedback loop is created which ensures that G2 phase commences until S-phase has been nearly completed (i). Under RS conditions, S-phase length is extended as a consequence of prolonged ATR-Chk1 signalling activation (ii). Loss of Cdc6 and Cdc7 functions leads to a failure of DNA replication licensing and the subsequent replication. This diminishes the activation of ATR pathway, leading to an accelerated progression of G2/M phases (iii). (*b*) ATR couples S-phase progression with G2/mitotic entry by inhibiting both CDK2 and CDK1 activity (left). This function is critical during replication stress (RS) conditions because cells must ensure full duplication and the genome prior to mitotic entry (right). During RS, increased ATR activity leads to inhibition of CDK2, which in turn downregulates global origin firing, prolonging the S-phase. Furthermore, ATR also delays the activation of the FOXM1-mediated mitotic gene network expression by inhibiting the CDK1 activity. Additionally, ATR prevents stalled fork collapse by regulating local origin firing pattern, although the mechanism is less well understood.

mitosis is of paramount importance for genome stability [16,27–32]. Thus, it is not surprising that RS is being recently recognized as a key predisposing factor for cancer development [33].

In this review, we will discuss the cellular responses and molecular pathways employed by cells to mitigate the negative effects of mild RS, with a focus on the emerging models of DNA repair mechanisms occurring post S/G2 phase, namely in early mitosis. Furthermore, we will review several new studies that have started to unravel the mechanistic insights of how RS induces chromosome instability (CIN), and how these lead to mitotic faults/defects.

# 2. The replication stress response

RS can be caused by a variety of endogenous and exogenous factors, which may elicit different cellular responses. Apart from the types of RS inducers, the timing and duration of RS are also important. For instance, if APH treatment is administered earlier in the cell cycle, i.e. in early S-phase compared with late-S/G2, it increases the severity of mitotic defects dramatically, presumably due to accumulated DNA damage and increased illegitimate DNA repair [34]. How these parameters affect cellular responses to RS in the context of intra-S-phase stages remains an important question, but for simplicity, we will restrict our review to the discussion of the effects of strong and mild RS on DNA replication and cell cycle progression.

During unchallenged DNA replication, the CMG helicase unwinds the double-stranded DNA at origins, leading to the initial formation of replication forks. DNA polymerases incorporate nucleotides on the leading and lagging strands—a process known as replication elongation. If the helicase and polymerase on the leading strand become uncoupled from one another, generally in face of RS, the rate of DNA synthesis is compromised. Under more severe conditions, the replication forks can even stall. Besides, a stalled fork can collapse into single-/double-stranded breaks if the stress is not relieved timely. Strong RS, therefore, is usually referred to as severe DNA damage in the form of single-/double-stranded breaks or DNA-DNA/protein-DNA crosslinking that cause acute stalling or collapse of replication forks. This can be introduced by ionizing- or UV-irradiation, radiomimetic drugs (e.g. neocarzinostatin and doxorubicin), topoisomerase poisons (e.g. camptothecin and etoposide) that covalently link topoisomerases to the DNA, and, as mentioned above, high doses of APH or HU treatments. Stalled/collapsed forks typically activate the intra-S-phase checkpoint to slow down S-phase progression for fork repair. The cells may arrest in G2 if the DNA breaks are persistent [35]. The detailed mechanism of how these checkpoints operate has been extensively reviewed elsewhere and are not the focus of this review [36].

Mild RS can be introduced through over-expression of proto-oncogenes (e.g. cyclin E) or experimentally by applying low doses of APH (a typical range from 0.1 to 0.75 μM) [24]. It is generally thought that mild RS causes retarded fork movements but allowing for continuous cell cycle

progression, albeit at a slower pace [37]. Mild RS treatments may mimic the intrinsic, pathological RS environment in cancerous cells, which do not trigger acute block on cell cycle progression [38]. So, it is commonly used as an important tool to study the link between RS and genome instability. For this reason, the rest of the review will mainly focus on the cellular responses to mild RS.

## 2.1. The effect of replication stress on genome-wise replication profiles

Extensive research has been done to understand how mild RS affects global genome duplication. Cells under mild RS, for instance, induced by low doses of APH, exhibit extended S-phase and a delay in G2 as well as mitotic entry. This presumably allows extra time to compensate for the slow DNA synthesis rate [14,34,37,39]. Under such conditions, local dormant origins are thought to be fired whereas global origin firing is suppressed [3,14,40]. This type of regulation potentially prevents unnecessary competition of replication resources, such as nucleotides, between the ongoing and the new firing origins. To understand the global effect of mild RS, a new study investigates the effect of mild RS on the temporal order of DNA replication across the bulk genome. Interestingly, they provide evidence to show that the replication timing of most genomic loci is largely unchanged and only 4% of them altered. Out of these affected regions, approximately two-thirds show a replication delay, namely that initiation or progression of DNA replication is postponed. Surprisingly, some regions display an opposite, advanced replication timing pattern. Nevertheless, because only 4% of the genome is affected, this indicates that cells remain in control of normal replication profiles throughout the vast majority of the genome even in the presence of mild RS [41]. However, the regions that are susceptible to replication timing changes mostly sat on common fragile sites (CFSs). Historically, CFSs have been identified as sites of fragility manifested as the formation of metaphase chromosome gaps/breaks and/or constrictions, usually following the treatment of low doses of APH [42]. The Glover laboratory has provided seminal work in furthering our understanding of CFS biology. They provide the first line of evidence that CFSs are under-replicated regions, whose stability requires DNA repair genes including FANCD2 and ATR-Chk1 [43–45]. CFSs exist across mammals [46], with over 100 APH-induced CFSs being identified in the human genome, their instability is cell-type and RS-inducer specific [47,48]. CFS instability has been correlated with a scarcity of replication origins, the presence of secondary structures that may be caused by the underlying AT-rich sequences and the collision between replication and transcription machinery [47,49–52]. A recent interesting study also shows that the strength of transcription activities on genes lying on CFSs vary the outcomes of CFS instability; namely, a weak promoter destabilizes large CFS genes whereas a strong one alleviates the instability [53]. Besides, late replication timing at CFSs has been previously proposed as a fragility factor [47,54]. Two independent groups, however, have recently argued against this proposal by showing that most CFSs, in fact, begin DNA synthesis in early/mid-, rather than late S-phase [41,55]. Therefore, these factors are not sufficient to determine CFSs on their own, implicating the complexity of the causes of CFS fragility. Indeed, Sarni *et al*. [41] propose that the fragility signature at CFSs is dependent on replication timing, active transcription and chromosome architecture. Specifically, CFSs are found at large genes that are actively transcribed and experience replication timing delay, and which are localized at topologically associated domain (TAD) boundaries. Because replication, transcription and DNA repair are coordinated within TADs, localization of CFSs at TAD boundaries was hypothesized to be the underlying fragile feature [41]. Nevertheless, because of such high recurrent instability, CFS expression is associated with pre-cancerous lesions and is believed to be an important driving factor during tumorigenesis [56].

## 2.2. The effect of replication stress on replication fork structures: uncoupling, remodelling and restart

Apart from the effect of replication timing, immense progress has also been made into understanding how replication fork structures are affected under RS at a molecular level. RS usually leads to the generation of stalled replication forks—a three-way replication intermediate structure [16]. The helicase and polymerase become uncoupled from one another, leading to the formation of a long tract of ssDNA (figure 2a). As a consequence of this uncoupling event, the rate of DNA synthesis is decreased and the fork fails to progress. It also triggers the intra-S-phase checkpoint via the ATR-CHK1 axis. If the stalled fork is not protected or resumed timely, it can collapse, and that increases

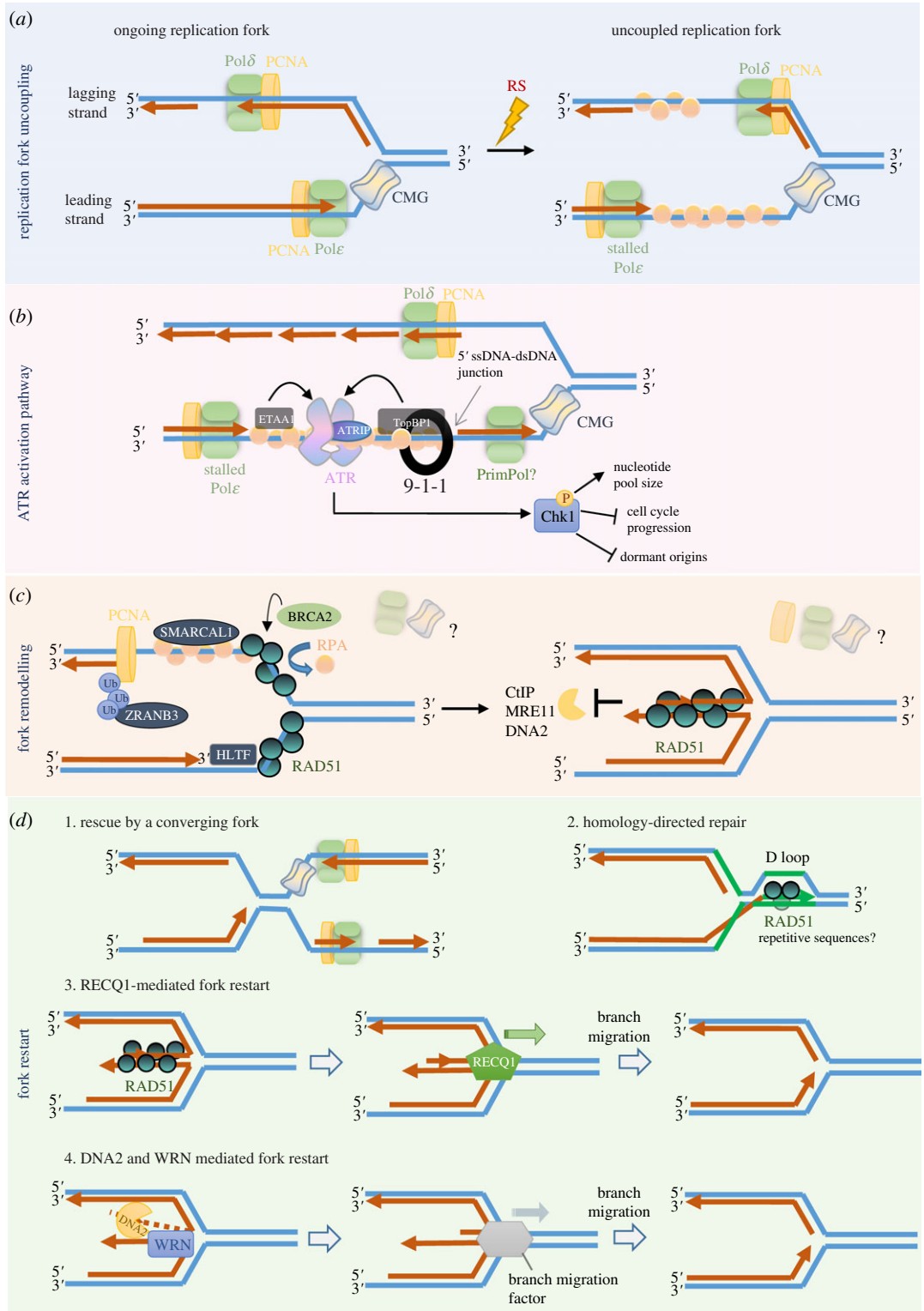

**Figure 2.** (*Caption opposite.*)

DNA damage and genomic rearrangements [33]. To guard genome integrity, cells have evolved a number of mechanisms to protect and restore DNA synthesis at the stalled forks.

Replication protein A (RPA) is believed to be one of the first responders to fork stalling. It acts by binding to the long ssDNA tracks behind the fork. Upon the coating of ssDNA with RPA, ATR and its partner ATRIP are recruited to the stalled forks [9,36]. While ATR/ATRIP's recruitment is via RPA, its activation requires TOPBP1 and ETAA1 [57–60]. TOPBP1 is simultaneously activated by the 9-1-1 checkpoint complex (RAD9-RAD1-HUS1), which associates to the 5′ ended ssDNA-dsDNA junctions

**Figure 2.** (*Opposite.*) The replication stress response. (*a*) During unperturbed S-phase, the CMG complex unwinds the DNA duplex for DNA synthesis mediated by two distinct DNA polymerases, Polδ and Polε, on the lagging and leading strand, respectively. PCNA acts as a processivity factor of both DNA polymerases. Following replication stress, the MCM helicase may uncouple from the DNA polymerase, leading to the generation of a long tract of ssDNA. RPA binds and stabilizes the ssDNA structure. Given the discontinuous synthesis of the lagging strand by Okazaki fragment formation, stalling Polδ may only lead to short stretches or gaps of ssDNA. On the other hand, uncoupling CMG from the Polε DNA polymerase on the lagging strand can generate a long region of ssDNA. The accumulation of bound RPA will then lead to the activation of ATR-CHK1 checkpoint signalling. (*b*) Upon the coating of the ssDNA by RPA at the stalled fork, ATR is recruited through its partner, ATRIP. To facilitate ATR signalling, two addition proteins, ETAA1 and TopBP1 are required. ETAA1 is recruited to the stalled fork through its interaction with RPA. By contrast, the activation of TopBP1 requires the presence of the Rad9-Rad1-Hus1 (9-1-1) clamp. Loading of the 9-1-1 clamp by the Rad17 clamp loader (not shown) requires the presence of a 5′ ssDNA-dsDNA junction, which is not normally present at a stalled leading strand. PrimPol can be recruited at the stalled fork and initiate primer synthesis, which creates a 5′ ssDNA-dsDNA junction. Successful activation of TopBP1 would lead to the subsequent activation of ATR. In turn, ATR phosphorylates its downstream partner, Chk1, leading to a cascade of checkpoint signalling and stalled fork repairing events. (*c*) Fork uncoupling also leads to fork remodelling processes. RAD51 replaces RPA on the ssDNA region in a BRCA2-dependent manner to stabilize the stalled fork. ZRANB3 is recruited following DNA-damage-induced ubiquitination of PCNA while SMARCAL1 is recruited via its direct interaction with RPA. HLTF binds the 3′ −OH group on the stalled leading strand. The binding of these fork remodelling factors mediates fork reversal, leading to the formation of a chicken foot structure via a yet unclear mechanism. The regressed arm of the fork becomes a single-ended double-stranded break and is prone to cleavage by nucleases (e.g. CtIP, MRE11, DNA2) but is protected by RAD51 in an enzymatic activity-independent manner. (*d*) Possible pathways to restart stalled forks. The regressed fork can be stabilized and protected from degradation until it is rescued by a nearby travelling converging fork (1). RAD51 can use its strange-exchange enzymatic activity to promote formation of a D-loop that promotes the re-initiation of replication, especially at regions of highly repetitive sequences (2). Regressed fork can also be restarted by the branch-migrating activity of RECQ1 to restore a three-way replication fork structure (3). Lastly, WRN2-DNA2 controlled resection can also promote fork restart, presumably through the recruitment of other branch migration factors, although the mechanism is not clear (4).

at the fork [6]. ETAA1 was also shown to be recruited by RPA to activate ATR [58–60] (figure 2*b*). The existence of two activating factors was proposed to enforce two independent and parallel ATR signalling pathways [61]. Given that TOPBP1 requires the presence of ssDNA-dsDNA junctions, which are more likely to form during RS, it was proposed to be the main activator of RS response, while ETAA1 might be required for, but not restricted to, regulation during unperturbed conditions [62]. Accordingly, ETAA1, but not TOPBP1, is required for ATR-mediated suppression of FOXM1 during unperturbed S-phase to regulate the S/G2 transition [15].

Under conditions of high RS, ATR prevents genomic instability through phosphorylation of Chk1 and together activates a number of downstream pathways. Some of those include regulation of RNR enzyme to control the nucleotide pool size, repression of DNA replication at dormant origins, and the delay of cell cycle progression through intra S-phase checkpoint activation [62,63]. In conditions of low RS, the phosphorylation of Chk1 is not always detectable, and is dependent on the duration of the stress. A short pulse (1–4 h) of sub-nanomolar concentrations of APH, although significantly reducing replication fork speeds, is not sufficient to cause strong Chk1 activation [37,64]. However, even if Chk1 is activated by prolonged treatment with APH, it nevertheless fails to stop mitotic onset [38]. It is probably because there is no severe DNA damage under mild RS to activate the G2/M DNA damage checkpoint [37].

Apart from activating the ATR signalling pathway, fork uncoupling has been associated with the recruitment of several fork remodellers (figure 2*c*). BRCA2 has been detected to bind to the ssDNA regions to displace bound RPA molecules, while assisting the loading of RAD51, a key initiator of homologous recombination [65]. At present, RAD51 has been shown to play at least two roles at the stalled forks. First, RAD51 forms nucleoprotein filaments on the ssDNA, which are stabilized by BRCA2, to protect the fork from nucleolytic degradation [66,67]. Second, RAD51 is capable of mediating fork reversal (also known as fork regression) independently of BRCA2, by converting the three-way junction of the replication fork into a four-way junction through a yet unknown mechanism [64,66,68]. This structure has also been commonly referred to as a 'chicken foot', where the fourth arm of the structure is generated through the annealing of the nascent lagging and leading DNA strands [69]. Fork reversal occurs in response to a wide variety of RS, such as DSB inducers (camptothecin, doxorubicin), SSB inducers (etoposide), DNA crosslinking reagents (mitomycin C, cisplatin) and DNA synthesis inhibitors (aphidicolin, hydroxyurea) [64]. It is also promoted by dsDNA translocases (ZRANB3, SMARCAL1, HLT) and DNA helicases (FANCM, FANCJ, WRN) [69–73]. The relationship

between these fork remodellers and RAD51, as well as their timing recruitment is highly complex. A complete review of the fork remodelling process can be found in [74]. The roles of fork regression remain an open question at present, but it is generally believed that by reversing back the extended chicken foot structure, it can bypass the lesion and resume DNA replication. This can also act as a fine-tuning system to control the appropriate speed of fork progression when DNA replication is challenged [75]. Another unanswered question is how ATR-mediated signalling is coupled with fork remodelling. While some studies show that ATR might counteract reversal by limiting the activity of SMARCAL1 [72], other studies show that fork reversal factors such as FANCM, FANCJ and WRN are required for ATR signalling (reviewed in [62]). Giving the plethora of signalling pathways that ATR mediates, at present, it remains a technical challenge to dissect the precise role of ATR on the remodelling of stalled forks spatio-temporally.

Regardless of whether forks have undergone the remodelling processes, fork restart mechanisms have been identified to occur at stalled forks (figure 2$d$). Not surprisingly, the factors involved depend on the type of RS, the combination of remodelling factors, and also whether the forks collapse due to spontaneous ssDNA or dsDNA breaks or programmed cleavages. Several mechanisms for stalled fork rescue have been well documented, and they include: DNA synthesis by the neighbouring origins or the nearby converging fork, strand invasion and D-loop formation by HR factors (BRCA1/2, RAD51, RAD52), remodelling at reversed forks by RECQ1, WRN2 and DNA2, or PrimPol-mediated repriming [75–78].

# 3. Mitotic DNA repair synthesis or break-induced mitosis DNA synthesis

## 3.1. The proposed mitotic DNA repair synthesis pathway

Unlike double-stranded DNA breaks, some forms of stalled or slowly progressing forks resulting from mild RS do not necessarily trigger a robust G2/M checkpoint to prevent cells from entering mitosis. As a result, cells proceed with chromosome segregation despite the presence of DNA replication intermediate structures. As mentioned above, this under-replication problem predominately arises at common fragile sites (CFSs). In order to preserve CFS integration, a recent study suggests that cells can initiate a process called mitotic DNA repair synthesis (MiDAS) in an attempt to complete DNA synthesis in early mitosis, and hence to eliminate late replication intermediates. MiDAS was claimed to be a type of break-induced DNA replication (BIR) process because known factors for BIR such as RAD52 and POLD3 are required [79–82]. To initiate BIR, the generation of a single-ended DSB substrate and a DNA template with homology sequences are needed. Similarly, it was proposed that MiDAS is initiated during chromosome condensation in prophase, where the stalled forks are exposed to cleavages by the MUS81-EME1-SLX4 complex that is activated by CDK1. This results in the formation of a single-ended DSB, which is then channelled to a RAD52-dependent strand annealing process on the sister chromatid. DNA (re)synthesis is subsequently carried out by a POLD3-containing polymerase Polδ. However, a recent study shows that depletion of either condensin I or condensin II, the essential complexes for chromosome condensation, does not impair, but increases MiDAS [83]. Therefore, whether DNA compaction is the trigger for MiDAS remains elusive. On the other hand, a study proposes that CDK1-mediated phosphorylation of RECQL5 at Ser727 is required for MiDAS initiation. It was shown that the phosphorylated RECQL5 removes the RAD51 filament coated on the ssDNA at the stalled forks. This then promotes fork breakages by recruiting the MUS81-EME1 endonuclease [81]. Therefore, the appearance of cytogenetic gap/break at CFSs is believed to be owing to the break-induced replication action instead of the conventional thought of the presence of incompletely replicated DNA structures. This claim is supported by the fact that cells exhibit reduction of CFS expression after MUS81 or POLD3 depletion [79]. In addition, the expression of CFSs has also been shown to reduce after inhibition of DNA polymerases during early mitosis [79]. While this indicates that the formation of chromosomal gaps/breaks may result from the unscheduled DNA synthesis activity during MiDAS, a recent report was not able to detect a similar suppression on CFS expression by high doses of aphidicolin, arguing against DNA synthesis activity in mitosis *per se* is one of the causal factors of CFS expression [83]. Recently, an elegant study using a cell-free *Xenopus* system showed that the breakage of stalled replication forks in mitosis is caused by the disassembly of replisomes. Deng and colleagues show that the increased CDK1 activity in mitosis activates TRAIP, an E3 ubiquitin ligase, which ubiquitinates the CMG helicase, leading to replisome dissociation and the subsequent fork breakages [84]. Perhaps the disassembly of the CMG complex might permit the access of structure-specific endonucleases such as MUS81 that breaks the stalled forks [82], which

normally is suppressed during normal DNA replication [85]. However, unexpectedly, Deng and colleagues also find that the mitotic fork breakage in *Xenopus* extracts is independent of MUS81 [84]. It is also independent of chromosome condensation. While this observation may not be consistent with the MiDAS pathway in human cells, it indicates that mitotic fork breakages could be potentially triggered by other structure-specific endonucleases, such as XPF/ERCC1 and SLX1/SLX4. However, if replisome disassembly is a prerequisite step for MiDAS initiation, it remains unclear why an increased loading of MCM complex was observed in mitosis following APH treatments [79].

Nevertheless, the generation of a single-end DSB may trigger a DNA damage response (DDR) even if cells have entered mitosis. Such structures are thought to be an ideal substrate for the RAD52-dependent BIR pathway [80]. Recently, a cryo-EM study has revealed new insights into the RAD52-mediated annealing mechanism; an important process of the microhomology search [86]. RAD52 has a ring-like structure possessing an outer and inner DNA-binding pocket. It contacts the ssDNA through its outer DNA-binding pocket leading to a trapped DNA-protein conformation that promotes further RAD52 accumulation. Then, the ssDNA will be moved into the inner DNA binding pocket, where it will be changed to a B-form conformation. Lastly, the ssDNA inside the ring can perform a homology search for ssDNA molecules bound by another RAD52 ring. When the homology search has been successful, the ssDNA will be released from the inner DNA pocket [86]. It is noteworthy that strand invasion and D-loop formation are required before the microhomology search. Whether RAD52 performs these roles during MiDAS or other factors are involved demands further investigation. Additionally, it was also claimed that the recruitment of MUS81 to CFSs is dependent of RAD52, although the mechanism is not fully understood [80]. While MiDAS is believed to be a RAD52-dependent BIR pathway, a recent report surprisingly shows that RAD52 is dispensable for mitotic DNA synthesis in non-cancerous cells. Instead, they find that FANCD2 plays a more important role among different cell lines [87]. It is, however, possible that cells may adapt a RAD52-independent BIR mechanism. Alternatively, it remains possible that some activity of mitotic DNA synthesis may be contributed by other DNA transactions and metabolic pathways. Indeed, the phenomenon of DNA repair synthesis during mitosis has been documented more than 50 years ago in cells treated with UV-irradiation [88]. Recently, it has also been demonstrated by using micro-focus laser [89]. However, in contrast to the replication-coupled 'MiDAS', mitotic DNA synthesis induced by UV- or microlaser-irradiation is not restrained within prophase but active throughout mitosis. In addition, it is worth pointing out that the core MiDAS factors mentioned above also play important roles during DNA replication and repair during S-phase; therefore, interferences of their interphase activities could potentially directly or indirectly affect MiDAS responses. A short summary of these functions is presented in table 1.

## 3.2. Unanswered questions of MiDAS

### 3.2.1. How are MiDAS promoting factors regulated temporally and spatially?

The discovery of MiDAS for the repair of RS-associated lesions is compelling and it raises many important questions. Increasing studies suggest that MiDAS is a highly complicated process involving multiple DNA replication and repair factors [79–82,87,105]. The TRAIP ubiquitin ligase has been proposed to facilitate MiDAS presumably via the activity of releasing replisomes that leads to fork breakages [82,84]. In S-phase, TRAIP also has a critical role in mediating the ATR signalling in response to replication interference (see below) [90]. Thus, defects in TRAIP increase RS. Intriguingly, while depletion of TRAIP diminishes MiDAS activity, it also diminishes the recruitment or association of FANCD2 on mitotic chromosomes [82]. FANCD2 has been shown to associate with CFSs in both interphase and mitotic cells under RS [27]. Functionally, it has been shown to protect the stalled forks from recession and/or to promote the resolution of DNA:RNA hybrid structures resulting from the collision between replication and transcription machinery [26,41,45,47,50,106,107]. Therefore, the observation of the loss of FANCD2 in TRAIP-depleted cells with increased RS is less expected. If FANCD2's loading at the stalled forks can occur in interphase where replisomes remain intact, it raises the question 'why the removal of replisome by TRAIP is required for its mitotic association?' Or, is it possible that TRAIP is another loading factor for FANCD2 during S/G2 phase? On the other hand, if replisome unloading is required for MiDAS initiation, then why do Minocherhomji *et al.* [79] detect more MCM complex on the mitotic chromosomes after RS? Is it possible there is a yet unknown reloading mechanism of MCM complex in human mitotic cells? Thus, it seems the coordination and regulation of the proposed MiDAS network proteins is highly complicated.

**Table 1.** A summary of other proposed roles of core MiDAS factors.

| MiDAS promoting factor | proposed roles in MiDAS | other cellular roles | references |
|---|---|---|---|
| TRAIP | disassembly of CMG to allow access of endonucleases | canonical DNA replication | [90] |
| | | ATR signalling during replication stress | [90] |
| | | spindle assembly checkpoint | [91] |
| MUS81 | endonuclease cleavage at replication forks to generate single-ended DSB | break-induced replication following fork collapse in interphase | [92] |
| | | MUS81-EME1-SLX4 is required for the resolution of single/double Holliday junctions | [93] |
| | | MUS81-EME1 promotes faithful disjunction of sister chromatids at CFSs | [94,95] |
| | | MUS81-EME2 cleaves *in vitro* reversed replication forks and D-loops | [96] |
| | | MUS81-EME2 is required *in vivo* for the restart of stalled forks in S-phase | [97,98] |
| | | alternative-lengthening of the telomeres | [99] |
| RAD52 | strand invasion/annealing to sister chromatids | regulatory role in fork reversal during replication stress | [100] |
| | | repair of under-replicated structures at 53BP1 bodies | [101] |
| | | break-induced replication following fork collapse in interphase | [92] |
| | | alternative-lengthening of the telomeres | [99] |
| POLD3 | elongation of DNA synthesis following strand annealing | lagging strand synthesis during S-phase as part of Pol$\delta$ complex | [102–104] |
| | | break-induced replication following fork collapse in interphase | [92] |

### 3.2.2. Is MiDAS a mitosis-driven process?

The simple answer is yes, as Minocherhomji *et al.* [79] have shown that the unscheduled DNA synthesis is detected only after cells enter mitosis. It was not detectable (or undetectable) in cells arrested in G2 phase. Unfortunately, it has never been tested if a similar DNA (re)synthesis might also occur if the RS is relieved during late G2. Otherwise, this would further strengthen the idea of MiDAS as a mitosis-driven process. Cells have evolved a number of repair pathways to deal with double-/singled-stranded DNA damage, fork stalling, fork collapses and breakages during S and even in G2. It is highly likely that some, if not all of these pathways, are employed to maintain DNA synthesis at CFSs under normal and RS conditions [44,45,50,108,109]. However, why do these pathways all fail to ensure CFS replication completion in interphase? In principle, there is no reason why BIR could not be initiated before mitosis to restart replication. Indeed, a study has shown that the RAD52-dependent BIR pathway also operates to rescue collapsed forks during S-phase [92]. So, why do mammalian cells not trigger BIR at CFSs until mitosis, where the DNA (re)synthesis needs to proceed under an environment where chromosomes are being actively compacted?—a condition probably not favoured

for efficient DNA replication. As mentioned above, one possible reason may be that interphase cells are incapable of evicting the stalled, non-processive replisomes to generate the single-ended DBS substrate for BIR. If that is the case, it would be interesting to know the underlying mechanism(s) of how cells maintain the integrity of stalled replisomes throughout S/G2 phases. Another possibility that may explain why MiDAS is only triggered at prophase is the need for high levels of CDK1 activity. Accordingly, MiDAS has been proposed to be dependent on the activation of RECQL5 and MUS81-EME1 in a CDK1-dependent manner [79,81]. Since MUS81-mediated cleavage is a key initiation step of MiDAS, and its activation is cell-cycle regulated, this may provide an explanation for the mitosis-specific activation of MiDAS. In essence, MUS81 has been proposed to serve at least two roles in response to RS, which depend on its binding partner and phosphorylation status. When partnering with EME2, MUS81 can cleave stalled forks caused by HU in S-phase, and promotes their restart [97]. Besides, MUS81 is also activated as cells progress towards the end of interphase, where another of its partners, EME1, is phosphorylated by high CDK1 activity, allowing the binding of MUS81/EME1 to SLX1-SLX4 complex for activation. The tight control of MUS81/EME1 activation is critical for genome stability because premature activation of CDK1 by inhibiting WEE1 has been shown to cause chromosome pulverization in a MUS81-dependent manner [85,110,111]. Other than cleaving replication structures, MUS81 complex also acts as a resolvase for Holliday junctions [112], critical for facilitating sister-chromatid disjunction in mitosis. MUS81 has also been implicated in other cellular transactions, such as BIR and alternative-lengthening of telomeres (ALT) [92,99]. Apart from CDK1-mediated activation, MUS81 can also be phosphorylated by CK2 at Ser87 in G2/M and is required to maintain genome stability following RS [113]. Thus, CK2 and CDK1 phosphorylation of MUS81 in G2/M might be pre-requisite for MiDAS initiation. However, it is also worth noting that most of the MiDAS studies apply a G2 synchronization strategy where CDK1 activity is chemically inhibited by the RO3306 inhibitor. Given the proposed important role of CDK1 for MiDAS, one cannot yet rule out that the same repair reaction might have begun prior to mitosis, namely in late G2, if normal CDK1 activity was not suppressed. Furthermore, CK2-mediated phosphorylation at Ser87 of MUS81 is also seen in late G2 cells, which may also imply that MUS81's activation could have initiated prior to mitotic onset, but further investigation is required [113].

### 3.2.3. What are other functions of the MiDAS promoting factors?

Another matter we would like to discuss in this review is the potential roles of MiDAS promoting factors for CFS stability. Depletion of MiDAS promoting factors POLD3, TRAIP and RAD52 has been associated with increased numbers of CFS-associated UFB structures in anaphase and 53BP1 nuclear bodies in offspring G1 cells [79,80,82]. Following this observation, it was proposed that the operation of MiDAS during prophase is critical to alleviating severe genome instability and segregation errors at CFSs. While this is very reasonable, we would also like to point out the other important roles of these factors before and during mitosis. It has been shown that POLD3 depletion leads to a concomitant loss of POLD1 subunit, which destabilizes the POLδ complex in interphase cells [114]. As a consequence, POLD3 haploinsufficiency in mouse B cells causes decreased fork speeds, which in principle could exacerbate the effect of RS, such as that induced by APH during S-phase. This subsequently could lead to more mitotic chromosomal lesions [114]. On the other hand, work from Hoffmann *et al.* [90] shows that TRAIP depletion in human cells leads to reduced fork velocity in unperturbed cells, which is exacerbated in the presence of HU treatment [90]. Additionally, TRAIP is important during the RS response as it associates with PCNA and mediates RPA coating of ssDNA, facilitating robust ATR signalling [90]. Furthermore, knockdown of TRAIP in S-phase, but not G2, leads to increased chromosome segregation defects, underlying its importance during scheduled DNA replication. TRAIP has also been shown to play a role in mitotic progression by regulating the spindle assembly checkpoint [91]. In parallel, RAD52 has also been shown to mediate fork stability in a pathway that is independent of MiDAS or DSB repair. Shortly after HU treatment, RAD52 is recruited at ssDNA where it forms a physical block against fork remodellers, such as SMARCAL1, preventing excessive fork reversal [100]. Depletion of RAD52 leads to excessive remodelling at the fork and exhaustion of the RAD51 pool, leaving long stretches of ssDNA unprotected, which will be degraded by exonucleases [100]. In addition to this, a very recent study also shows that 53BP1 bodies repair the under-replicated DNA structures using a RAD52-dependent pathway in late S-phase, G2 and sometimes prophase [101]. Therefore, the contribution of MiDAS promoting factors for CFS protection is likely to be through multiple pathways and throughout the cell cycle. Thus, to dissect their specific role in MiDAS will require more sophisticated analysis.

### 3.2.4. Is BIR efficient to complete DNA replication in mitosis?

MiDAS is proposed to be the 'last resort' mechanism to complete genome duplication before sister chromatids disjunction. Peculiarly, MiDAS is tightly restrained within (early) prophase, and does not occur after cells reach the (pro)metaphase stage. Given the very short period of prophase (typically less than 30 min), one would expect that the efficiency of MiDAS might be extremely high, perhaps faster than normal DNA replication. However, early studies from yeasts have shown that BIR involves multiple rounds of strand invasion and dissociation. Besides, the Pol32/POLD3-containing Polδ polymerase was shown to be prone to slippage or stalling during synthesis [115]. This unstable D-loop dissociation and reformation increase the chances of template switching, which leads to toxic genomic rearrangements [115]. However, whether this D-loop instability is merely a result of studying a chromosome fragmentation vector, or also presents itself when sister-chromatid templates are used requires further investigation. Apart from that, the initiation of new DNA synthesis by Polδ during BIR was also shown to be a rate-limiting step, which seems to require hours to repair the single-ended DSBs [116]. Recently, another yeast study also reveals that the migration of D-loops is limited by Mus81-mediated resolution activity in order to suppress the high mutagenicity nature of BIR. This is suggested to allow a more faithful DNA synthesis carried out by the neighbouring converging fork [117] (figure 3e (i)(ii)). Since most CFSs are located within chromosomal arms and not near telomeres [47], the completion of DNA synthesis between the stalled converging forks in mitosis would probably require two BIR reaction that fills the gap in between them (figure 3e, iii). Collectively, under the context of active DNA condensation during prophase, it is hard to conceive how MiDAS can efficiently finish the business of 'incomplete DNA replication' in time, unless the under-replication regions are relatively small. However, a recent EdU-seq mapping study shows that the MiDAS regions contain a gap zone ranged from 500 kilobases to 1.2 Megabases [118]. Interestingly, Macheret *et al.* also detect a double-peak within a MiDAS region, approximately 40% of which resides at known CFSs. CFSs are well known to be of poor replication origin [55]. It thus implies that the formation of the double MiDAS peaks is a result of converging forks travelling along from the flanking origins. Alternatively, there may be two difficult-to-replicate sites within a single CFS zone. However, given that a higher dose of aphidicolin can further widen the peak-to-peak distances [118], this would rather support the former notion that two forks are heading towards each other, meeting at the central unreplicated region. MiDAS has shown to confine to prophase, which only lasts for approximately 30 min. Given that the average speed of the elongating replication fork in S-phase is approximately $1.6 \text{ kb min}^{-1}$ [119], it will need at least a few hours to fill up some gaps containing several kilobases. During S-phase, cells can solve this problem through using extra dormant origins. However, this seems not applicable to MiDAS regions, therefore whether large under-replication gaps can be filled up in time is debatable. It is possible though that there may be a yet unknown mechanism in mammalian cells to relieve all these limitations during mitosis.

Other than the proposed BIR activity, one cannot exclude that the phenomenon of unscheduled DNA synthesis in mitosis is a consequence of the resumed activity from the replicative DNA polymerases when the stress is relieved. In principle, there remains a short period of time in which the stalled replisome could carry out residual replication before its disassembly, given that the inhibitory action of APH is reversible. As mentioned above, a recent report demonstrates that the phenomenon of mitotic DNA synthesis is not necessarily dependent on RAD52. In addition, we also failed to detect a significant reduction in mitotic DNA synthesis in MUS81-knockout human cells following APH treatments (C Mocanu, M Fernández-Casañas, A Herbert, T Olukoga, ME Ozgurses, K-L Chan 2021, unpublished data). Thus, further investigation will need to clarify these discrepancies. Nevertheless, it is also interesting to ask whether MiDAS is a 'programmed' pathway for DNA replication completion, or if it is a 'bystander effect' where cells attempt to repair accidentally broken forks in mitosis, which preferential breakages are for the urgency of sister chromatids separation.

# 4. Mild replication stress induces structural and numerical chromosome instability and mutations

In the absence of MUS81-mediated cleavages, the under-replicated DNA structures impair proper sister chromatids disjunction, which can give rise to bulky and ultrafine anaphase bridges (UFBs) and lagging chromatin [28,31,94,95]. Consequently, the equal distribution of genetic content fails, and that also leads to micronucleation and aneuploidy [28,94]. Next, we will discuss several recent studies that have begun to expand our knowledge of the implications of RS on mitotic chromosomal abnormality.

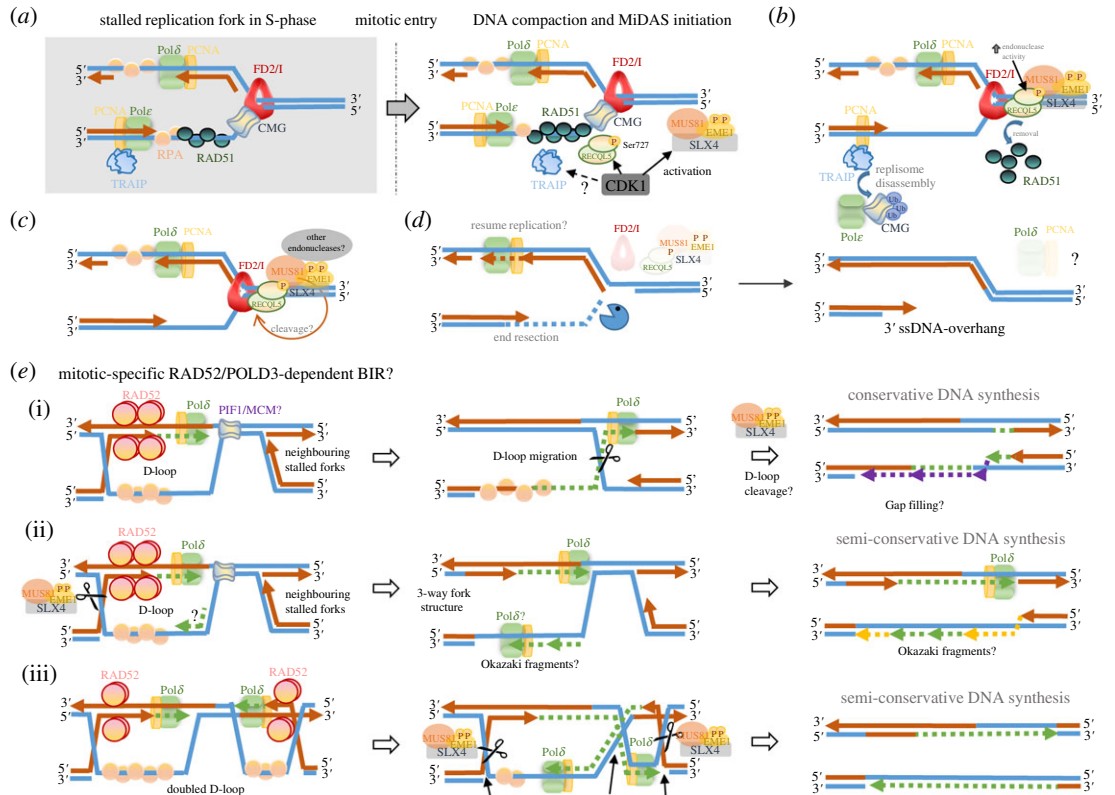

**Figure 3.** The MiDAS/break-induced mitotic DNA repair synthesis model. (*a*) Under mild replication stress (RS), cells enter mitosis with stalled forks at common fragile sites (CFSs) predominantly. The uncoupling of polymerases causes ssDNA accumulation, leading to RPA and RAD51 filament formation and the binding of FANCD2/I dimer, presumably to protect and stabilize the stalled forks. Upon mitotic entry, CDK1 phosphorylates multiple MiDAS promoting factors including RECQL5 helicase, EME1 (a partner of MUS81 endonuclease) and TRAIP. (*b*) TRAIP poly-ubiquitinates MCM helicase, driving replisome disassembly. In parallel, the activated MUS81-EME1 is recruited probably via the scaffold platform of SLX4. The phosphorylated RECQL5 removes the RAD51 filaments from the leading strand. Furthermore, RECQL5 can also physically associate with MUS81 that promotes its endonuclease activity. (*c*) Consequently, MUS81-EME1 probably cleaves the leading strand of the stalled fork and generates a single-ended DSB. (*d*) To generate a 3′ ssDNA overhang for homologous strand invasion, the cleaved arm is probably undergoing a 5′ to 3′ resection by an exonuclease(s). (*e*) RAD52 mediates the strand annealing step followed by the formation of D-loop. A POLD3-containing polymerase then resynthesizes DNA at the incompletely replicated regions ahead. Helicases like PIF1 or MCM may need to facilitate the DNA synthesis progression. (i) The D-loop can migrate with the POLD3-containing polymerase that may hit a neighbouring stalled fork. As a result, a Holliday junction (HJ)-like structure is formed. The gap of the nascent synthesized strand may be filled (the purple dotted line) during D-loop migration or after the HJ resolution, resulting in a conservative DNA synthesis. (ii) If the D-loop is resolved immediately after the stand invasion, it may restore a three-way replication fork structure for DNA resynthesis in a semi-conservative manner. (iii) Alternatively, the stalled converging forks can both initiate BIR, leading to the formation of a double D-loop. DNA resynthesis between the D-loops will fill up the under-replication gap region and generate dsDNA catenane. The single or double D-loop may then be cleaved by MUS81-EME1 and/or other structural-specific endonucleases to resolve sister-DNA entanglements.

One of the main drivers of genomic instability during cancer development is the loss of maintenance of chromosome integrity, also termed as CIN. CIN can be manifested as structural and/or numerical changes on chromosomes. It is also frequently associated with CFSs, regions of the genome prone to rearrange and/or mutate in cells experiencing RS. Both Rosselli's and Hickson's laboratories have shown that the absence of MUS81-EME1 mediated cleavage at CFSs leads to increased mis-segregation errors [94,95]. It was later also proposed as the loss of MiDAS activity that leads to the accumulation of stalled fork structures. In anaphase, they manifest as a distinct class of UFBs known as fragile site UFBs (fs-UFBs) while sister chromatids are pulled apart. This particular class is distinct from the centromeric UFBs (c-UFBs) and telemetric UFBs (t-UFBs) as the fs-UFBs are usually associated with FANCD2 and FANCI at its termini [27,28,120]. The BTRR complex (BLM, Topoisomerase 3A, RMI1 and RMI2) and PICH are found to decorate along the DNA bridges. One proposed role of the PICH/BTRR complex is to promote

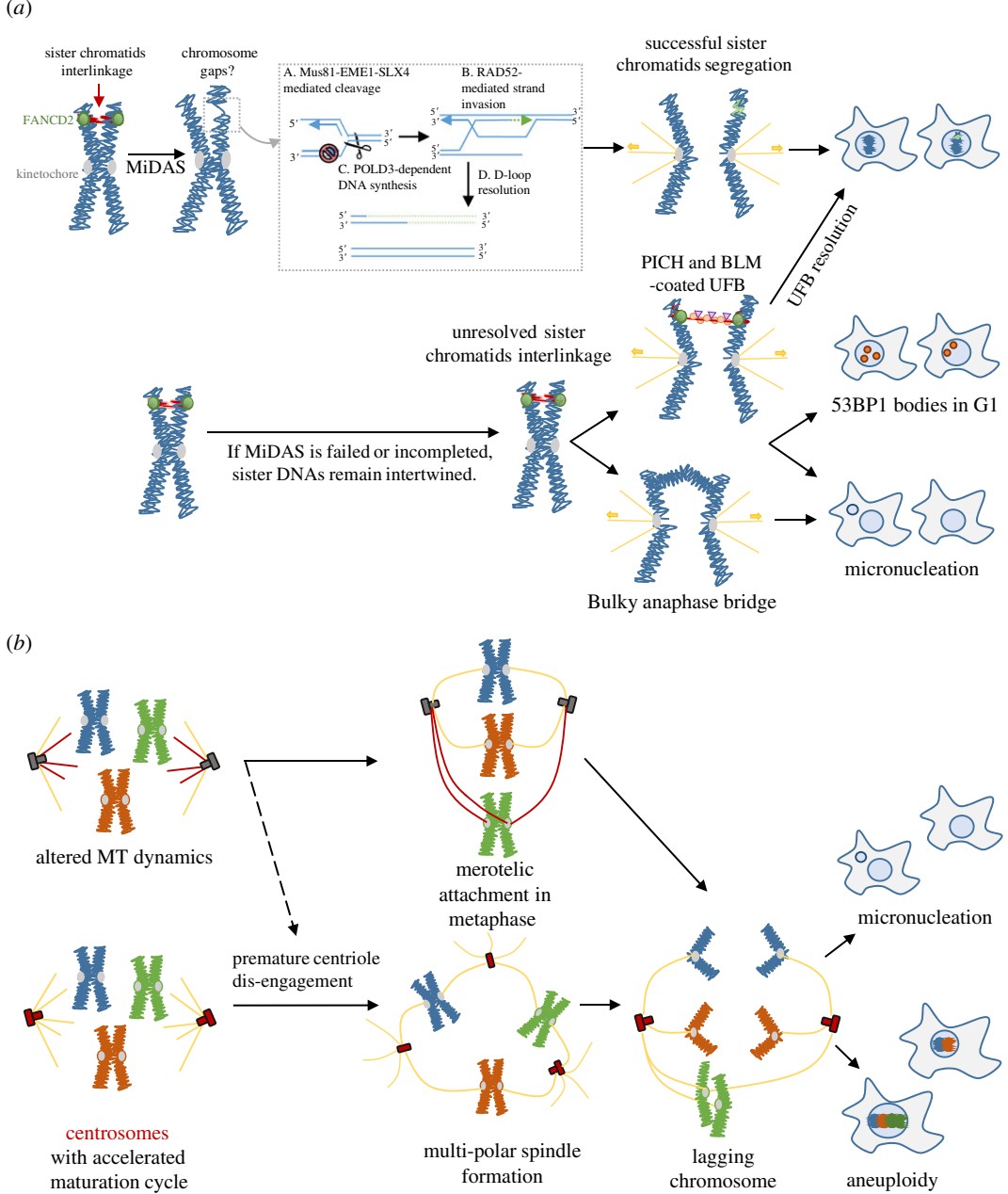

**Figure 4.** Mild replication stress leads to chromosome segregation defects that increase genome instability. (*a*) Cells can commit mitosis despite the presence of under-replicated DNA structures under mild replication stress conditions. This leads to the interlinkage of sister chromatids. FANCD2/I dimer and MUS81-EME1-SLX4 complex are recruited to these structures, presumably to protect or resolve the DNA linkages. This has been suggested to cause the cytogenetic breaks and/or gaps on the mitosis chromosomes. The stalled fork breakage due to the MUS81-EME1 cleavage was proposed to initiate MiDAS/BIR reaction. If MiDAS is completed, it will convert the stalled fork structure to recombination structures such as D-loop or HJ-like molecules. They may be resolved during prophase/prometaphase or, otherwise, can generate ultrafine DNA bridges in anaphase cells, which may be resolved by the PICH/BLM complex. If MiDAS is defective or incomplete, sister chromatids remain intertwined by the replication intermediates that again can form UFBs and/or bulky chromatin bridges during anaphase. The failure or inappropriate resolution of the replication intermediates can lead to chromosome non-disjunction and/or chromatin damage, resulting in micronuclei formation or 53BP1 nuclear bodies in the newborn G1 daughter cells. (*b*) Mild RS has also been shown to increase aneuploidy as it can alter microtubule (MT) spindle dynamics and induces premature centriole disengagement. This presumably promotes erroneous merotelic-kinetochore attachments in metaphase, which can lead to lagging chromosome formation during anaphase. Alternatively, premature centriole disengagement can cause multi-polar spindles formation and chromosome mis-segregation. It was hypothesized that the centrosomes have accelerated the maturation cycle as a result of prolonged S/G2 delay induced by replication stress. Under these conditions, it increases the chances of whole chromosome mis-segregation.

resolution of the interlinking DNAs [121–124]. Another recent study suggests that the complex may also trigger DNA de-compaction at the surrounding entangled sister chromatids to facilitate their separation [120,125,126]. The failure or inappropriate resolution of the DNA bridges can undoubtedly cause DNA damage. Two independent groups have detected that following mild RS, cells show increased formation of 53BP1 nuclear bodies at CFSs in offspring G1 cells [29,127]. 53BP1 focal formation is part of the DDR, thus implicating that under-replicated structures can generate DNA/chromatin lesions in the descendant cell population [127] (figure 4a). Therefore, one mechanism leading to the structural change of chromosomes by RS is likely to be attributed to the post-mitotic damage generated during the chromosome segregation process at CFSs.

A number of reports also indicate that RS leads to lagging chromosome formation and numerical CIN [128–130]. However, until recently, the mechanism linking RS with numerical CIN was elusive. One recent study has provided new insights into this matter. Wilhelm and colleagues offer convincing evidence that the increases of numerical CIN in a variety of human cell lines, including untransformed and cancerous cells by RS are a result of premature centriole disengagement and transient multipolar spindle formation [129]. Following the observations that centrioles split immediately after prometaphase entry, the authors propose that the disengagement is not due to prolonged mitotic arrest. Instead, they find that this is mediated by the activities of ATR and CDK1 in G2. Because RS generally leads to a delay in G2 progression, the increased time in G2 period is likely to cause the premature activation of centrosome splitting [129]. In addition, they also demonstrate that microtubule dynamics is altered and that may be partially responsible for promoting premature centriole disengagement. Indeed, increasing evidence shows that microtubule dynamics are altered under RS conditions [34,38]. One group has found that whole chromosome mis-segregation in non-transformed cells under mild RS is due to increased microtubule growth rates, which might cause merotelic microtubule-kinetochore attachments [38]. This observation was also reproducible in CIN cancer cells, indicating that RS-mediated microtubule dynamics change is linked with genomic instability (figure 4b).

Apart from numerical CIN, a report shows that, in colorectal cancer cells, 70% of the chromosome segregation errors are due to RS-induced structural CIN, visualized as the formation of acentric chromosome and bulky anaphase bridges [30]. Undoubtedly, lagging chromatin or chromosome can cause aneuploidy. More importantly, they also lead to chromothripsis, a situation where the laggard DNAs are shattered into thousands of clustered fragments, being rearranged and integrated into the daughter cell genomes [131]. A recent analysis of chromothripsis signatures across thousands of tumour genomes reveals a pervasive feature and suggests chromothripsis is one of the major driving factors for cancer genome evolution [132]. Structural CIN usually results in gross chromosome rearrangements including chromosome fusions, translocations, deletions and amplifications. They also commonly arise at CFSs (full review in [47]), but the underlying cause(s) have not been unified and it is highly likely to be a multifaceted process. Regions around CFSs are found to have high mutation rates. Given that MiDAS is proposed to be an error-prone BIR pathway, this provides a reasonable explanation for the sequence instability at CFSs. However, DNA lesions at CFSs are also evident in interphase, especially in late S and G2. Thus, it is also possible that some mutations are acquired prior to mitosis via other damage and repair pathways, such as translesion DNA synthesis (TLS) [108,133–135]. So, a number of theories, which are not mutually exclusive, have been put forward to try to explain the high CIN rates at CFSs. They include: the use of high error-prone pathways (e.g. MiDAS, TLS), condensation- and/or endonuclease-mediated chromatin damage, improper resolution of sister-DNA linkage structures, and whole chromosome mis-segregation [27,28,79,94,95,127,129].

# 5. Concluding remarks

More evidence has emerged that there is a tight connection and coordination among cell cycle progression, DNA replication and mitosis. Undoubtedly, the proceeding of chromosome condensation and segregation on DNA molecules that contain replication and/or recombination intermediate structures are devastating to genome integrity [27,136,137]. So, why can cells not delay mitosis under mild RS conditions until whole genome duplication is complete? Some possible speculations that are not mutually exclusive include: (i) The delayed transition to and/or the arrest of G2/M may be dependent on the quantity, scale, conformation of the stalled forks, and even the locations of the under-replication regions. The structure of a slowly progressing replication fork may not always be sufficient to trigger and maintain a checkpoint pathway as compared with the completely stalled or

collapsed forks. (ii) Alternatively, cells may overcome G2 delay, if it is present, as prolongment of G2 phase is also detrimental to the cell. It can lead to centromere over-duplication, resulting in multipolar spindle formation and whole chromosome mis-segregation [138,139]. (iii) Delaying mitosis onset might have no use to restart the stalled forks if cells require a 'programmed cleavage and repair' that only occurs in mitosis. In conclusion, RS can affect genome integrity in multiple cell-cycle stages. How and to what extent cells can achieve DNA synthesis completion and repair under such challenges would be the next interesting question for scientists.

Data accessibility. This article has no additional data.

Authors' contributions. C.M. and K.-L.C. designed the content of and wrote the manuscript. C.M. helped draft the manuscript and designed the figures. K.-L.C. revised and edited the manuscript substantially, and modified the figures. All authors gave final approval for publication.

Competing interests. The authors declare no competing financial interests.

Funding. The research was supported by Sir Henry Dale Fellowship to K.-L.C. by Wellcome Trust and the Royal Society (104178/Z/14/A).

Acknowledgements. We thank members in the Chan laboratory and Ulrich Rass for helpful discussion. We also thank Tomisin Olukoga for proofreading this review. K.-L.C is supported by Sir Henry Dale Fellowship provided by Wellcome Trust and the Royal Society.

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
