## [Peer Review File · Royal Society Open Science]

Review History

RSOS-201932.R0 (Original submission)

Review form: Reviewer 1

Is the manuscript scientifically sound in its present form?

Yes

Are the interpretations and conclusions justified by the results?

Yes

Is the language acceptable?

Yes

Do you have any ethical concerns with this paper?

No

Have you any concerns about statistical analyses in this paper?

No

Recommendation?

Accept as is

Comments to the Author(s)

This is a timely, insightful and up-to-date review focusing on recent advances in understanding the processes that shield our genomes from the consequences of replication stress. It is great to see the mechanistic aspects of MIDAS and the role of cell cycle regulation discussed in depth. In addition, the figures provided are extremely informative.

Below are just a few suggestions for the authors:

Page 6, line 31: Authors suggest that replication stress inducing drugs, such as HU and Aphidicolin, are more harmful in early than late S-phase. This is a very interesting observation and it would be helpful to provide a reference.

Page 10, line 41: Authors state that a recent study has shown that condensin depletion increases MIDAS, however a reference is not provided.

Page 10, lines 55-60: Authors report the conflicting findings of two studies: one showing that inhibiting polymerases during MIDAS reduces CFS expression and one finding no difference. It may be useful to discuss the fact that both studies inhibit actual synthesis using a high dosage of aphidicolin, but potentially the events preceding MIDAS are still going on. Additionally, one of the studies, reference 32 is cited as a pre-print; this is now available as a peer-reviewed paper in Cell Reports in case the authors prefer to update the reference.

Page 11, line 41: Authors provide an interesting discussion on whether MIDAS is a mitotic-driven process and if so, how is this controlled. They provide two mechanisms, one via eliciting replication forks and one based on CDK1 dependency. It may be interesting to discuss the cell cycle-dependent regulation of Mus81 as another potential regulatory mechanism confining MIDAS to mitosis.

Page 14, line 52: Authors refer to a study using EdU-seq to map MIDAS using high-throughput sequencing, but unfortunately a reference is not provided. I believe the authors are referring to the paper by Macheret et al in Cell Research, 2020. Since this paper represents a recent and significant advance in understanding MIDAS, it may be useful to add the reference and expand on the results of that study.

Review form: Reviewer 2

Is the manuscript scientifically sound in its present form?

Yes

Are the interpretations and conclusions justified by the results?

Yes

Is the language acceptable?

Yes

Do you have any ethical concerns with this paper?

No

Have you any concerns about statistical analyses in this paper?

No

Recommendation?

Accept with minor revision (please list in comments)

Comments to the Author(s)

This submission reviews the cellular response to the presence of under-replicated DNA, including break-induced mitotic replication and the induction of chromosomal instability by mild

replication stress. The subject is timely, the coverage is comprehensive and provides an overview of the main issues that currently drive research in the field. Overall, this is a very clear review that will be useful to the field. Below are some minor suggestions for further improvement.

1. In addition to Figure 1, the description of sequential kinase-mediated events that regulate cell cycle progression would benefit from a flow chart or another form of visualization.
2. It would be helpful to define, explicitly, at first use, the basic concepts of mild and strong replication stress. Before any discussion of the cellular response, it would be useful to define molecular characteristics of each state (mild vs strong) and mention the main characteristics of the cellular responses to mild and strong stress.
3. The discussion of other functions of Midas-promoting factors is basically a list of functions attributed to specific mediators. A table would have been helpful

Minor

1. Why are CFS and R-loop sometimes referred to as "so called" CFS and R-loops? These are acceptable terms. It might be better to define them on the first use and continue to use them without the qualifier.
2. Page 3 line 55 "inhibiting CDC6" might be misleading, because inhibition is usually a term referring to enzymatic activity.

Decision letter (RSOS-201932.R0)

Dear Dr Chan

On behalf of the Editors, we are pleased to inform you that your Manuscript RSOS-201932 "Mind the replication gap" has been accepted for publication in Royal Society Open Science subject to minor revision in accordance with the referees' reports. Please find the referees' comments along with any feedback from the Editors below my signature.

Please submit your revised manuscript and required files (see below) no later than 7 days from today's (ie 28-Apr-2021) date. Note: the ScholarOne system will 'lock' if submission of the revision is attempted 7 or more days after the deadline. If you do not think you will be able to meet this deadline please contact the editorial office immediately.

Please note article processing charges apply to papers accepted for publication in Royal Society Open Science (<https://royalsocietypublishing.org/rsos/charges>). Charges will also apply to papers transferred to the journal from other Royal Society Publishing journals, as well as papers submitted as part of our collaboration with the Royal Society of Chemistry

(<https://royalsocietypublishing.org/rsos/chemistry>). Fee waivers are available but must be requested when you submit your revision (<https://royalsocietypublishing.org/rsos/waivers>).

on behalf of Dr Ed Bolt (Associate Editor) and Malcolm White (Subject Editor)
openscience@royalsociety.org

Associate Editor Comments to Author (Dr Ed Bolt):

Associate Editor: 1

Comments to the Author:

Please respond to the comments from one of the reviewers, who suggests making some changes in a revised version of the manuscript.

Reviewer comments to Author:

Reviewer: 1

Comments to the Author(s)

This is a timely, insightful and up-to-date review focusing on recent advances in understanding the processes that shield our genomes from the consequences of replication stress. It is great to see the mechanistic aspects of MIDAS and the role of cell cycle regulation discussed in depth. In addition, the figures provided are extremely informative.

Below are just a few suggestions for the authors:

Page 6, line 31: Authors suggest that replication stress inducing drugs, such as HU and Aphidicolin, are more harmful in early than late S-phase. This is a very interesting observation and it would be helpful to provide a reference.

Page 10, line 41: Authors state that a recent study has shown that condensin depletion increases MIDAS, however a reference is not provided.

Page 10, lines 55-60: Authors report the conflicting findings of two studies: one showing that inhibiting polymerases during MIDAS reduces CFS expression and one finding no difference. It may be useful to discuss the fact that both studies inhibit actual synthesis using a high dosage of aphidicolin, but potentially the events preceding MIDAS are still going on. Additionally, one of the studies, reference 32 is cited as a pre-print; this is now available as a peer-reviewed paper in Cell Reports in case the authors prefer to update the reference.

Page 11, line 41: Authors provide an interesting discussion on whether MIDAS is a mitotic-driven process and if so, how is this controlled. They provide two mechanisms, one via eliciting replication forks and one based on CDK1 dependency. It may be interesting to discuss the cell cycle-dependent regulation of Mus81 as another potential regulatory mechanism confining MIDAS to mitosis.

Page 14, line 52: Authors refer to a study using EdU-seq to map MIDAS using high-throughput sequencing, but unfortunately a reference is not provided. I believe the authors are referring to the paper by Macheret et al in Cell Research, 2020. Since this paper represents a recent and significant advance in understanding MIDAS, it may be useful to add the reference and expand on the results of that study.

Reviewer: 2

Comments to the Author(s)

This submission reviews the cellular response to the presence of under-replicated DNA, including break-induced mitotic replication and the induction of chromosomal instability by mild replication stress. The subject is timely, the coverage is comprehensive and provides an overview of the main issues that currently drive research in the field. Overall, this is a very clear review that will be useful to the field. Below are some minor suggestions for further improvement.

1. In addition to Figure 1, the description of sequential kinase-mediated events that regulate cell cycle progression would benefit from a flow chart or another form of visualization.
2. It would be helpful to define, explicitly, at first use, the basic concepts of mild and strong replication stress. Before any discussion of the cellular response, it would be useful to define molecular characteristics of each state (mild vs strong) and mention the main characteristics of the cellular responses to mild and strong stress.
3. The discussion of other functions of Midas-promoting factors is basically a list of functions attributed to specific mediators. A table would have been helpful

Minor

1. Why are CFS and R-loop sometimes referred to as "so called" CFS and R-loops? These are acceptable terms. It might be better to define them on the first use and continue to use them without the qualifier.
2. Page 3 line 55 "inhibiting CDC6" might be misleading, because inhibition is usually a term referring to enzymatic activity.

===PREPARING YOUR MANUSCRIPT===

If you have been asked to revise the written English in your submission as a condition of publication, you must do so, and you are expected to provide evidence that you have received language editing support. The journal would prefer that you use a professional language editing

service and provide a certificate of editing, but a signed letter from a colleague who is a native speaker of English is acceptable. Note the journal has arranged a number of discounts for authors using professional language editing services (<https://royalsociety.org/journals/authors/benefits/language-editing/>).

===PREPARING YOUR REVISION IN SCHOLARONE===

-- If you have uploaded ESM files, please ensure you follow the guidance at <https://royalsociety.org/journals/authors/author-guidelines/#supplementary-material> to

include a suitable title and informative caption. An example of appropriate titling and captioning may be found at https://figshare.com/articles/Table_S2_from_Is_there_a_trade-off_between_peak_performance_and_performance_breadth_across_temperatures_for_aerobic_sc_ope_in_teleost_fishes_/3843624.

Author's Response to Decision Letter for (RSOS-201932.R0)

See Appendix A.

Decision letter (RSOS-201932.R1)

Dear Dr Chan,

I am pleased to inform you that your manuscript entitled "Mind the replication gap" is now accepted for publication in Royal Society Open Science.

You can expect to receive a proof of your article in the near future. Please contact the editorial office (openscience@royalsociety.org) and the production office (openscience_proofs@royalsociety.org) to let us know if you are likely to be away from e-mail contact – if you are going to be away, please nominate a co-author (if available) to manage the proofing process, and ensure they are copied into your email to the journal. Due to rapid publication and an extremely tight schedule, if comments are not received, your paper may experience a delay in publication.

Kind regards,
Royal Society Open Science Editorial Office

on behalf of Dr Ed Bolt (Associate Editor) and Malcolm White (Subject Editor)
openscience@royalsociety.org

Appendix A

Responses to reviewers' comments

We thank you both reviewers for taking their time to go through our review and for their very constructive and helpful comments. We have now amended our manuscript accordingly.

Reviewer comments to Author:

Reviewer: 1

Comments to the Author(s)

This is a timely, insightful and up-to-date review focusing on recent advances in understanding the processes that shield our genomes from the consequences of replication stress. It is great to see the mechanistic aspects of MIDAS and the role of cell cycle regulation discussed in depth. In addition, the figures provided are extremely informative.

Below are just a few suggestions for the authors:

Page 6, line 31: Authors suggest that replication stress inducing drugs, such as HU and Aphidicolin, are more harmful in early than late S-phase. This is a very interesting observation and it would be helpful to provide a reference.

We have corrected the sentence. The harmfulness is determined by the duration of the drug treatments, thus early S phase rather late S phase cells probably accumulate more damage.

Page 10, line 41: Authors state that a recent study has shown that condensin depletion increases MIDAS, however a reference is not provided.

Added

Page 10, lines 55-60: Authors report the conflicting findings of two studies: one showing that inhibiting polymerases during MIDAS reduces CFS expression and one finding no difference. It may be useful to discuss the fact that both studies inhibit actual synthesis using a high dosage of aphidicolin, but potentially the events preceding MIDAS are still going on. Additionally, one of the studies, reference 32 is cited as a pre-print; this is now available as a peer-reviewed paper in Cell Reports in case the authors prefer to update the reference.

Added

Page 11, line 41: Authors provide an interesting discussion on whether MIDAS is a mitotic-driven process and if so, how is this controlled. They provide two mechanisms, one via eliciting replication forks and one based on CDK1 dependency. It may be interesting to discuss the cell cycle-dependent regulation of Mus81 as another potential regulatory mechanism confining MIDAS to mitosis.

We have added the relevant discussion on pages 12 and 13.

Page 14, line 52: Authors refer to a study using EdU-seq to map MIDAS using high-throughput sequencing, but unfortunately a reference is not provided. I believe the authors are referring to the paper by Macheret et al in Cell Research, 2020. Since this paper represents a recent and significant advance in understanding MIDAS, it may be useful to add the reference and expand on the results of that study.

We have added the reference and expanded our discussion on this study.

Reviewer: 2

Comments to the Author(s)

This submission reviews the cellular response to the presence of under-replicated DNA, including break-induced mitotic replication and the induction of chromosomal instability by mild replication stress. The subject is timely, the coverage is comprehensive and provides an overview of the main issues that currently drive research in the field. Overall, this is a very clear review that will be useful to the field. Below are some minor suggestions for further improvement.

1. In addition to Figure 1, the description of sequential kinase-mediated events that regulate cell cycle progression would benefit from a flow chart or another form of visualization.

We have made a new Figure 1 according to the suggestion.

2. It would be helpful to define, explicitly, at first use, the basic concepts of mild and strong replication stress. Before any discussion of the cellular response, it would be useful to define molecular characteristics of each state (mild vs strong) and mention the main characteristics of the cellular responses to mild and strong stress.

We have now added the description to define strong and mild replication stress on pages 5 and 6.

3. The discussion of other functions of Midas-promoting factors is basically a list of functions attributed to specific mediators. A table would have been helpful

A table has been added.

Minor

1. Why are CFS and R-loop sometimes referred to as “so called” CFS and R-loops? These are acceptable terms. It might be better to define them on the first use and continue to use them without the qualifier.

We have deleted “so-called” and defined CFSs and R-loops in the text.

2. Page 3 line 55 “inhibiting CDC6” might be misleading, because inhibition is usually a term referring to enzymatic activity.

We have changed the sentence (...by conditional degradation of CDC6 and chemical inhibition of CDC7...).